# A Fast and Efficient Ultrasound-Assisted Extraction of Tocopherols in Cow Milk Followed by HPLC Determination

**DOI:** 10.3390/molecules26154645

**Published:** 2021-07-30

**Authors:** Archimede Rotondo, Giovanna Loredana La Torre, Teresa Gervasi, Giacomo di Matteo, Mattia Spano, Cinzia Ingallina, Andrea Salvo

**Affiliations:** 1Department of Biomedical and Dental Science and Morpho-Functional Imaging (BIOMORF), University of Messina, Polo Universitario Annunziata, 98168 Messina, Italy; arotondo@unime.it (A.R.); tgervasi@unime.it (T.G.); 2Department of Drug Chemistry and Technologies, University of Rome “La Sapienza”, P.le Aldo Moro, 5, 00185 Roma, Italy; giacomo.dimatteo@uniroma1.it (G.d.M.); mattia.spano@uniroma1.it (M.S.); cinzia.ingallina@uniroma1.it (C.I.)

**Keywords:** tocopherol, vitamin E, milk, food analysis, HPLC/FD analysis

## Abstract

A fast HPLC method with fluorescence detector (FD) was developed for the determination of three tocopherols (TOCs) in milk samples from *Modicana* cattle breed. The ultrasound-assisted procedure was optimized for the extraction of TOCs prior to HPLC/FD analysis, reducing sample preparation time and allowing a fast quantification of α-tocopherol, δ-tocopherol and γ tocopherol. The optimized ultrasonic extraction combines an efficient and simple saponification at room temperature and a rapid HPLC quantification of TOCs in milk. The precision of the full analytical procedure was satisfactory and the recoveries at three spiked levels were between 95.3% and 87.8%. The linear correlations were evaluated (R^2^ > 0.99) and the relative standard deviation (RSD) values for intra-day and inter-day tests at three spiked levels were below 1% for the retention time and below 5.20% for the area at low level spiking. The proposed procedure, reducing the experimental complexity, allowed accurate extraction and detection of three TOCs in milk samples from *Modicana* cattle breed.

## 1. Introduction

Tocopherols (TOCs), together with tocotrienols, are fat-soluble compounds that make up the group of vitamin E. Among these, α-tocopherol (α-TC) is the most studied, as its biological activity is well demonstrated and assessed; for this reason, especially in the nutritional field, often the name “tocopherol” is used in the singular, as synonymous with vitamin E. In nature there are four tocopherols; α-TC is the “most active” one and it is used to compare the activity of the other tocopherols: β-tocopherol (β-TC), γ-tocopherol (γ-TC), δ-tocopherol (δ-TC) [1]. From the structural point of view, the TOCs can be considered tocol (6-hydroxy-chroman) derivatives, bearing a methyl group in position 2 and an aliphatic side chain with 16 carbon atoms (phytol). The four TOCs share the same hydroxy-chromic core connected to the saturated carbon chain and differ to each other by the methylated positions over the aromatic ring [2].

It is a general belief that TOCs exert strong antioxidant activities both in vivo and in vitro. Evidence from several studies suggest that TOCs in vivo: (a) may protect lipids from oxidation; (b) may preserve biological membranes by tackling the peroxidation of polyunsaturated fatty acids triggered by free radicals generated inside the cells [3,4]; and, (c) may play a vital role in preventing atherosclerosis, cardiovascular disease and free radical damage to cells, responsible for the pathological changes associated with aging [5]. It has also been seen that these vitamins promote the synthesis of heme and hemic proteins (hemoglobin, myoglobin, cytochrome, peroxidase, and so on) [6,7], prevents renal damage and chronic diseases associated with oxidative stress [8,9,10], play an immune role by increasing resistance to bacterial infection and have a protective action against cancer by the stimulation of the immune system [2,5,11].

In vitro TOCs are used as antioxidant additive and studies have mainly focused on the comparison of the antioxidant activities of α-TC and γ-TC [12]. Despite the general agreement that α-TC is the most efficient antioxidant in vivo, there was always a considerable discrepancy in its “absolute” and “relative” antioxidant effectiveness in vitro, especially when compared with γ-TC [3,13].

TOCs, as vitamins, are organic compounds that cannot be synthesized by the human organisms; therefore, they need to be introduced in limited quantities through food to reach the Recommended Daily Allowances (RDAs about 15 mg/day for adults) [14]. Tocopherols are mainly found in foods of vegetal origin; the richest sources are germ fraction of cereal grains, wheat germs, green leafy vegetables, oil seeds and fruits and their respective oils. In smaller quantities they are also present in foods of animal origin such as fish, meat, liver, eggs, milk, and dairy products [1].

In particular, the milk composition and the tocopherol content have often been the subject of several scientific investigations from 1948 [15] up to the present [16]. It was evidenced that the concentrations of TOCs in milk are dependent on animal species, breed, nature of forage fed to ruminant and stage of lactation [17]; moreover, to improve the milk quality and to maintain the oxidative stability of milk, TOCs were often used as additives in the diet of dairy cows [18,19].

The direct determination of TOC in milk is troublesome as it is a fat-containing food rich in triglycerides, interfering hydrophobic compounds, that can be co-extracted. One of the first strategies related to the α-tocopherol determination in cow milk involved a tricky colorimetric method [20] elaborated by modifying the Quaife method for the chemical assay of food for vitamin E content [21], and based on the colorimetric oxidimetric methods that use the reducing properties of tocopherol towards ferric chloride [22].

According to the literature data [19,23,24,25,26], the determination of TOC in milk was carried out through a complex workflow including many steps such as saponification, extraction, fractionation, and separation of the compounds using chromatographic techniques. Pretreatment steps are always necessary to extract and purify TOCs before further analyses, especially when analyzing TOCs by reversed-phase liquid chromatography (RP-HPLC), in which the separation of tocopherols from interfering compounds such as triglycerides is indispensable [23,24]. Particularly, several alkaline saponification procedures for the extraction of TOCs from milk have been described [19], but it was observed that under the alkaline hydrolysis conditions in a heated bath, TOCs are prone to decomposition, and then significant losses may occur [24,25,26]. Often, to avoid oxidation of TOCs, aliquots of antioxidants such as ascorbic acid [19,24] are added to the mixture before saponification. Moreover, after saponification, a liquid-liquid extraction with organic solvent is required, and often the solvent must be removed by rotavapor [27] or under nitrogen flow [19], involving relatively high volumes of organic solvent and time-consuming sample pretreatment. Because the procedures are complex and involve multiple-steps, the recovery and the accuracy of the results could be negatively influenced.

HPLC is the most widely used chromatographic technique employed to analyze TOCs, and both normal-phase (NP-HPLC) and reversed-phase (RP-HPLC) were applied [28]. In NP-HPLC tocopherols are separated using a relatively non-polar organic solvent and better selectivity was achieved using hexane with 1,4-dioxane (4–5%, *v*/*v*) as the polar modifier [28]. RP-HPLC has often been preferred for separation of TOCs because the chromatographic conditions are mainly based on less hazardous mobile phases, e.g., methanol or ethanol; nevertheless, this procedure requires fatty acid removal by saponification, to prevent the contamination of the RP column. The detection, identification, and quantification of TOCs in cow milk were usually performed by chromatographic procedures taking advantage from the use of photodiode array detector (PAD) or UV detector [17,27]; nonetheless, it produced weak signal-to-noise ratios and, consequently, was suitable just for tocopherol-rich samples. Better sensitivity was achieved by HPLC separation runs coupled with fluorescence detection for quantification of TOCs [19,25,29] or with mass spectrometry [23,30].

Gas chromatography (GC) could also be used to analyze TOCs [31], but that might include a risk for decomposition owing to high temperatures. Recently, near-infrared (NIR) and mid-infrared (MIR) spectroscopy has been proposed to predict the vitamin content in cow milk [16]. Certainly, this is an interesting and different approach for the determination of TOC overcoming difficulties associated with multi-stage methods. Unfortunately, by this strategy it is generally difficult to obtain high prediction quality when the milk compounds are of a low concentration [16].

On these premises, the aim of the study was the development of a simple, fast and efficient method for the simultaneous determination of α-TC, δ-TC and γ-TC in milk samples. The ultrasound-assisted extraction was used enabling a smart work up with good repeatability, and the limited time per sample paves the way to a high-throughput procedure deserving future developments. The TOCs levels were detected by NP-HPLC, and the mobile phase was optimized with respect to short time analysis and maximizing chromatographic resolution. Subsequently, the developed validated method was applied to determine the TOCs content in 36 milk samples from *Modicana* cattle breed, a local breed reared in Sicily that produce high quality milk.

## 2. Results and Discussion

### 2.1. Analytical Methods

The extraction procedure of TOCs from cow’s milk was optimized by performing different tests on a milk sample whose TOCs have been previously determined by the Havemose et al. [29] method.

The first step of this study regarded the optimization of the analytical parameters; therefore, initial trials to seek for the suitable solvent were performed. At the beginning the extraction efficiency of different solvents (i.e., *n*-heptane, *n*-hexane, and *n*-pentane) and sample to solvent ratios were evaluated also tuning the KOH/methanolic volume and concentration. As *n*-hexane is an efficient solvent for extracting lipophilic substances, as it is commonly available in any laboratory, it was selected as the extracting agent. The influence of the solvent volume evidenced that the highest recovery yield was obtained if the milk-solvent ratio was 1:1. If more *n*-hexane was added, higher volumes diluted the sample, affecting the detection of TOCs and leading to poor recoveries. During the sample preparation the milk samples were spiked with saponification solution containing different concentrations of methanolic KOH solution. The higher saponification efficiencies were obtained with 2 mL KOH 2M.

At first, the milk saponification was carried out by adding ascorbic acid (2 mL, 1% in methanol), as it is reported that TOCs are quickly degraded by light (mainly by the UV radiation), and that they are slowly oxidized by atmospheric oxygen (especially under heat and alkaline conditions) [25]. Preliminary tests showed that, for short treatment times, there were no significant differences between saponification with or without ascorbic acid. According to these experimental data, in order to simplify a shared routine procedure, ascorbic acid was not added for milk saponification.

The ultrasound extraction trials were run in a bath with fixed frequency (50/60 Hz) and power (200 W); therefore, the optimization of the ultrasonic conditions regarded the temperature and the extraction time. It is well known that temperature affects both solubility and stability; this is especially true for vitamins such as TOCs. As already mentioned, the alkaline hydrolysis conditions in a heated bath accelerate the decomposition of TOCs and often significant losses may occur, therefore several extraction tests were performed at constant temperatures of 10, 20, 30, and 40 °C.

Generally speaking, low temperature allowed a good recovery of all TOCs while higher values had a negative effect on the analyte’s extraction. Because 10 °C extraction did not allow a full recovery of TOCs, the 20 °C set was chosen for further experiments. At higher temperature, lower recoveries for the TOCs were recorded. To study the effect of the ultrasound time on the extraction efficiency, the sonication time was varied from 5 to 30 min. At 10 min, the higher extraction efficiencies were obtained for all TOCs; significant response drops were observed for longer sonication times and this may be because TOCs are prone to decomposition, and then remarkable losses may occur. Profiles obtained on TOCs for lower extraction time showed that all the analytes never reached the equilibrium condition (particularly α-TC), so significant peak decrement and lack of repeatability were observed.

In order to define the most suitable ultrasonic treatment, the evaluation of simultaneous variation of two extraction conditions, namely extraction temperature and sonication time, was specifically performed through the analytical technique applied by Derrnger and Suich [32]. This method converts several experimental responses (yi) into desirability functions (di(yi)), ranging from 0 to 1, approaching 1 as the response reaches its best target value [33]. The general desirability function D is the geometric mean of the different desirability factors (Equation (1)):(1)$$D={\left(\overset{n}{\underset{i=1}{\prod }}d_{i}\left(y_{i}\right)\right)}^{1/n}$$

In this case we have considered as responses the three peak areas, their deviation among triplicate measurements and recovery values [34]. Figure 1 displays the general desirability surface by changing the extraction conditions, so that it is possible to infer the best condition to run the analysis. Consequently, 20 °C and 10 min were selected as optimum temperature and time for the ultrasonic saponification reaction.

In the HPLC/FD analysis the normal phase was employed thanks to its direct compatibility with lipophilic analytes [19,28,29] and to reduce the sample preparation; the fluorescence excitation and emission wavelength (295 and 330 nm, respectively) were selected as previously established [19]. The mobile phase was optimized to achieve quick runs and maximum chromatographic resolution for each TOC. We compared the effects of different elution phases and chromatographic conditions on the base line, the peak shape, the selectivity, and the retention time of TOCs. The best conditions were accomplished using *n*-hexane/ethyl acetate (9:1, *v*/*v*) in isocratic elution with a flow rate of 0.8 mL min^−1^. The selected isocratic chromatographic conditions allowed a fast determination of TOCs, and all the analytes were separated with good resolution in 6.59 min (Figure 2). In order to evaluate the matrix effect and test the goodness of the chromatographic separation, three milk samples were initially tested to determine the content of α-TC, δ-TC, and γ-TC, then they were spiked with different increasing amounts of the TOC standard examined. Each TOC was added separately, and the milk sample was spiked with the standard analyzed distinctly to establish the exact retention time. The evaluation of these chromatograms showed that the peak area of α-TC, δ-TC and γ-TC gradually increased, and no interference was observed (Figure 2).

The column temperature was the last parameter tested in order to achieve better chromatographic resolution. Analyses carried out at 35 °C, 40 °C and 45 °C revealed that k’ values and the area versus temperature did not improve significantly, although above 45 °C there was a slight drift toward the area decrease. At the same time, as more symmetrical peaks with slightly higher areas were observed at 40 °C, the analyses were carried out at this temperature.

The method was validated following a specific protocol set up according to the requirements of the Commission Decision 2002/657/EC [35], and international guidelines [36] that our laboratory has now adopted for at least ten years [33,37,38]. The parameters determined were linear range, limit of detection (LOD) and limit of quantification (LOQ), accuracy, precision and short- and long-term repeatability. The analytical characteristics of the method are presented in Table 1 and Table 2.

For quantification purposes of TOCs, the external calibration procedure was carried out; therefore, the peak areas and the retention times of TOCs were compared with calibration curves freshly prepared each day before the measurements. To determine the linear range, for α-TC and γ-TC the five points calibration curves were drawn in the range 0.20–10.00 mg L^−1^, while for δ-TC linearity was achieved within the range 0.10–5.00 mg L^−1^. Each solution was injected six times and the linear coefficients, evaluated by the least square regression coefficients (R^2^), were higher than 0.998 for all the analytes. The compound identification was based on the retention time of each target compound.

The estimation of the limits of detection (LODs) and of the limits of quantification (LOQs) were calculated according to IUPAC guidelines. The LODs and the LOQs were experimentally calculated as 3.3 σ/S and 10 σ/S, respectively, where σ is the residual standard deviation and S is the slope of the regression curve. LOD values ranged between 0.027 and 0.030 mg L^−1^, while LOQ values were between 0.089 and 0.101 mg L^−1^, and the values obtained are really very close for all TOCs.

In an attempt to determine the accuracy in the validation study, the recovery test was conducted. In particular, the assessment was carried out by spiking a known amount of each TOC to a milk sample whose TOC content has been previously determined. The recovery study of the full analytical procedure was performed on a milk sample with a low content of TOCs and was carried out for three spiked levels of the mixed reference standards (0.5 mg L^−1^, 2.5 mg L^−1^, and 5 mg L^−1^, respectively) to the real sample. After addition, the whole analytical procedure was repeated three times at each concentration level. The obtained results are presented in Table 1, which shows satisfactory recovery values. As the data shows, the lowest recovery values were obtained for a low spiking level and these values ranged from 88.4% to 90.5% with the relative standard deviations (RSD%) not higher than 7.4%; on the other hands, the recovery results for a high spiking level showed higher values that ranged from 91.6% to 95.3% with RSD values lower than 0.6%.

To evaluate the precision of the method, the intra-day and the inter-day repeatability of retention time and peak area, with relative standard deviation (RSD%) were determined. The intra-day repeatability was estimated on the RSD% of the measurements obtained by performing six consecutive injections of TOC standard at three different concentrations (low, medium and high) under the selected conditions. The same standards were also analyzed over a period of twelve successive days to determine the inter-day RSDs. These data are listed in Table 2, and the analytical precision, assessed by means of the restricted repeatability and the intermediate repeatability, evidenced that for intra-day repeatability the peak area RSD values varied from 1.89% to 2.55% for low levels, and from 0.35% to 1.35% for high levels. The inter-day repeatability RSD values ranged from 3.95% to 5.12% for high levels and from 3.17% to 5.20 for low levels. Retention time RSD% values for intra-day repeatability (*n* = 6) were lower than 1% and ranged from 1.04 to 1.40 for inter-day repeatability (*n* = 12).

All these data evidenced that the method proposed was simple and suited to the goal we set out to pursue with this study; moreover, the recovery values determined were comparable to those obtained with the Havermose et al. [29] method, which consists in a saponification of the milk sample at 70 °C with saturated potassium hydroxide.

### 2.2. Analysis of Milk Samples

This optimized method was applied to determine the content of TOCs in 36 milk samples from *Modicana* cattle breed, a local breed reared in Sicily. These bovines are less productive than modern breeds and are at great risk of extinction today. Particularly, in 1985 *Modicana* was included into the Registry of autochthonous bovine populations and ethnic groups with limited diffusion, threatened with extinction, in order to safeguard the bovine breeds bred in Italy representing an unmissable genetic heritage [39]. *Modicana* cow farms keep modest sized herds and are generally managed under the traditional system based on pasture [40].

The milk samples were provided by three different producers from the Iblean area of Ragusa (Sicily, Italy); specifically, 11 samples were from *Farm A*, 12 from *Farm B*, and 13 from *Farm C*. Measured TOCs resulting from the milk samples of three different *Modicana* local cow farms are shown in Table 3 and graphically represented in Figure 3. It illustrates the range, the average value with the standard deviation and the median value determined for the samples in each farm.

The data, on the levels of TOCs determined, evidenced that all the analytes were consistently found above the LOQ values, with concentrations ranging from 3.46 mg L^−1^ (detected in milk sample from *Farm C*) to 7.79 mg L^−1^ (detected in milk sample from *Farm B*) for α-TC, from 2.62 mg L^−1^ to 4.01 mg L^−1^ (both detected in milk samples from *Farm B*) for γ-TC, and from 0.99 mg L^−1^ to 1.71 mg L^−1^ (both detected in milk samples from *Farm B*) for δ-TC. The mean values and the standard deviation together with the median values related to the 36 *Modicana* milk samples evidenced that there are no substantial variations from one sample to another, especially for sample from *Farm A*, and α-TC had the more significant levels in all the samples, although it was slightly higher in several samples from *Farm B* and *Farm C*. Only in two samples from *Farm C* were the α-TC contents comparable with those of γ-TC while δ-TC, with its lowest average content, presented a more homogeneous and uniform distribution among all the 36 *Modicana* cow milks. As shown in Figure 3, milk samples from *Farm B* had on average the maximum total content of TOCs (maximum level 12.72 mg L^−1^) and, in three samples from *Farm C,* the total level of TOCs was above 10.13 mg L^−1^, while in all the samples from *Farm A,* the total level of TOCs was shown to be at minimum (from 8.03 to 8.98 mg L^−1^). Compared with all the other milk samples, those from *Farm A* had a much more homogeneous distribution of TOCs, and this aspect could probably be the result of a variety of different factors. In fact, it is attested that TOCs are involved in nutritional milk’s distinctiveness and their concentrations and relative compositions are often related to genetic and physiological factors, animal’s forage diet, metabolism by the animal tissues, and environmental factors [17,41].

Although the trend of the reciprocal relationship among the concentrations of the TOCs determined reflects what was previously observed for other cow milk samples by other authors, and places α-TC as the predominant tocopherol and δ-TC as the least represented; nevertheless, the concentrations we determined in our samples were not similar to previous studies [19]. Particularly, *Modicana* milk samples reached considerably higher values of TOCs with respect to the other available studies [19,42]. To the best of our knowledge, this is the first study in which α-TC, δ-TC and γ-TC have been simultaneously determined in milk from Modicana cattle breed. The unique previous reference reporting the quantification of α-TC in milk from *Modicana* cattle breed [43], highlights that this milk contains very high levels of this metabolite, and the values were two to five times higher than other milk samples [19,42]. In this paper Guardiano observed that the rather higher levels of α-TC in *Modicana* milks were also maintained without cattle grazing. According to this finding [43], the differences in vitamin content in *Modicana* milk compared with other cows could be linked to genetic reasons and could derive from the ability of the *Modicana* cows to hold a large reserve of TOCs in fat storage tissues.

## 3. Materials and Methods

### 3.1. Chemicals, Reagents, and Samples

Methanol, *n*-hexane, and ethyl acetate were Optima UHPLC/MS and were purchased from Fisher Chemical products (Milan, Italy). Tocopherols (α-TC, δ-TC, and γ-TC) standard 98%, ascorbic acid, and KOH were obtained from Sigma-Aldrich (Milan, Italy). Stock standard solutions of TOCs were prepared in *n*-hexane (1000 mg/L) and quantification of each tocopherol was carried out according to an external calibration procedure. In particular, five-points calibration curves were constructed by serially diluting a stock solution of each commercial standards. Before HPLC analysis each solution was filtered through a non-sterile OlimPeak nylon syringe filter of 0.2 μm, purchased from Teknokroma (Monza Brianza, Italy). All measurements were conducted in triplicate.

Three *Modicana* local cow farms were involved in the present study, enrolled in the milk-recording agency of Ragusa (Sicily, Italy), supported by the *Sicilian Regional Agriculture and Forestry Department*, and managed under the typical regional conditions. The samples were collected from mid-March until the end of April, from 36 lactating cows. All the cows were managed according to free-stall housing, following the Directive 2010/63/EU on the protection of animal used for scientific purposes [44] and the Italian Regulation on animal care [45], and they showed no sign of illness during the experimental timespan. Milk samples were collected using hand milking, by standardized procedures at the same time of the day; each sample was immediately refrigerated on ice within 5–10 min; then, all the samples were frozen at 40 °C and stored until the final test. Before the analysis, each sample was warmed at room temperature.

### 3.2. Sample Pre-Treatment

Aliquots of 2 mL of cow milk were poured into a 10 mL dark screw-cup vessel (to protect TOCs from potential oxidation), with 2 mL *n*-hexane added, capped, and mixed for 10 s at room temperature. Then, 2 mL of methanolic KOH (2 N) were further added and the oxygen was removed from the vessel by purging the air space with nitrogen. Immediately, the vessel was capped and mixed for 10 s at room temperature again. The final solution was sonicated for 10 min at 20 °C. For the ultrasound extraction trials, a LBS1 10Lt (Falk Instruments, Milan, Italy) water bath with fixed frequency (50/60 Hz) and power (200 W) was used. To improve the efficiency of the extraction process, some parameters such as ultrasonic time, ultrasonic bath temperature and milk-liquid ratio were optimized. After the optimized ultrasonic conditions, the solution was centrifuged (4000× *g*, 10 min, +4 °C), the supernatant was filtered through a 0.2 μm OlimPeak syringe filter with a nylon membrane (Teknokroma, Monza Brianza, Italy), transferred into an amber vial, and analyzed by HPLC/FD.

### 3.3. HPLC Analysis

The TOCs analyses were carried out using an HPLC system (Shimadzu, Milan, Italy) equipped with a CBM-20A controller, a CTO-20A column oven, one LC-20AD pump, a DGU-20A3 degasser, and a RF-20A fluorescence detector. The data were processed with the software LabSolutions ver. 5.10.153 (Shimadzu). In all analyses, a *LiChrosorb^®^ Si 60* (5 µm) column (4.6 mm I.D. × 250 mm), protected by a guard column with the same stationary phase, was used. Analyses were run at 40 °C, under isocratic condition, with a mobile phase composed of *n*-hexane/ethyl acetate (90:10 *v*/*v*). The injection volume was 20 µL and the flow rate was 0.8 mL min^−1^. Fluorescence excitation and emission wavelength were 295 nm and 330 nm, respectively. A-TC, γ-TC, and δ-TC were detected and quantified at 4.67 min, 5.58 min, and 6.59 min, respectively (Figure 2). The quantitative analysis was carried out with the method of external standard using suitable calibration curves and any quantification was estimated as the mean value of three repeated measurements.

## 4. Conclusions

A rapid, efficient, and reliable method for the determination of TOCs (α, δ, and γ) in cow milk samples was developed. The proposed method, using ultrasound-assisted extraction, combines an efficient and simple saponification protocol at room temperature and a rapid HPLC quantification of TOCs in milk. Compared with other procedures, the sample pre-treatment procedure, based on the use of ultrasound with *n*-hexane at room temperature, enables shorter extraction times, prevents the thermal degradation of TOCs and allows a quick and reliable analytical process that, bypassing the addition of antioxidants, repeated extraction phases, and the evaporation to dryness of the extracting solvent, makes it possible to determine three TOCs with accurate and trustworthy detection results.

Compared with previous methods on milk sample, our proposed procedure allows accurate extraction and detection of TOC, reduces the experimental complexity, facilitates automation, and is thus suitable to routine work with lots of samples.

Based on this method, the concentrations of the target TOCs were estimated in milk samples from *Modicana* cattle breed, and the data evidenced that the milks contained significant amounts of all the analytes investigated. The present study provides new information on the content of three TOCs in milk from the peculiar *Modicana* cattle breed. These data suggest carrying out additional comparative studies on more samples, collected in different seasons and after specific cattle pasture, in order to better clarify the impact of the management system on the characteristics of milk from *Modicana* cows.

## Figures and Tables

**Figure 1 molecules-26-04645-f001:**
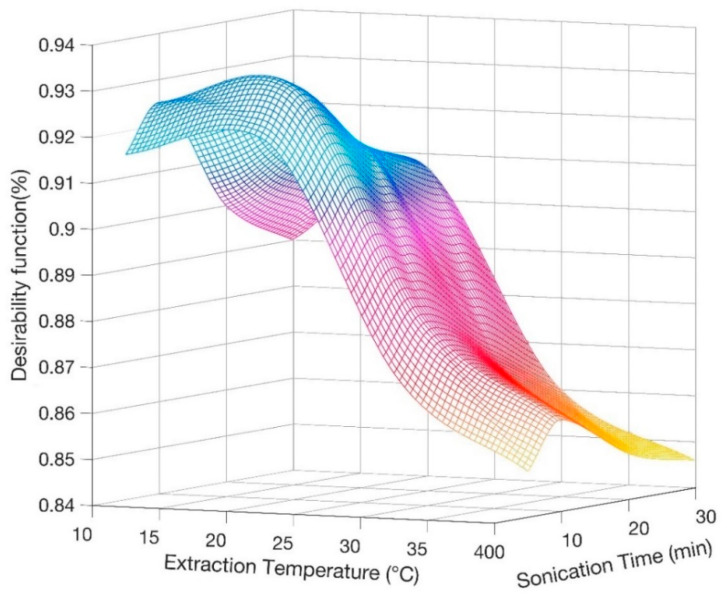
Derringer desirability function calculated as a function of extraction temperature and sonication time for average recovery of tocopherols (TOCs).

**Figure 2 molecules-26-04645-f002:**
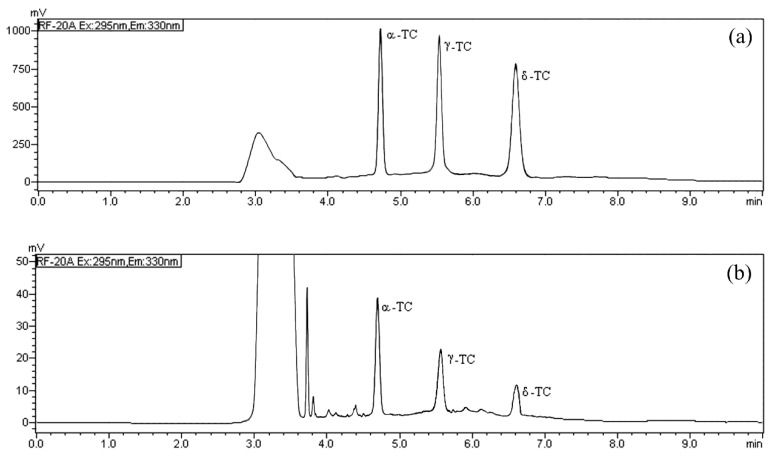
HPLC/FD chromatograms of: (**a**) α-tocopherol (α-TC), δ-tocopherol (δ-TC) and γ tocopherol (γ-TC) standard solution (10 mg L^−1^); and (**b**) of a real milk sample from *Modicana* cattle breed after extraction.

**Figure 3 molecules-26-04645-f003:**
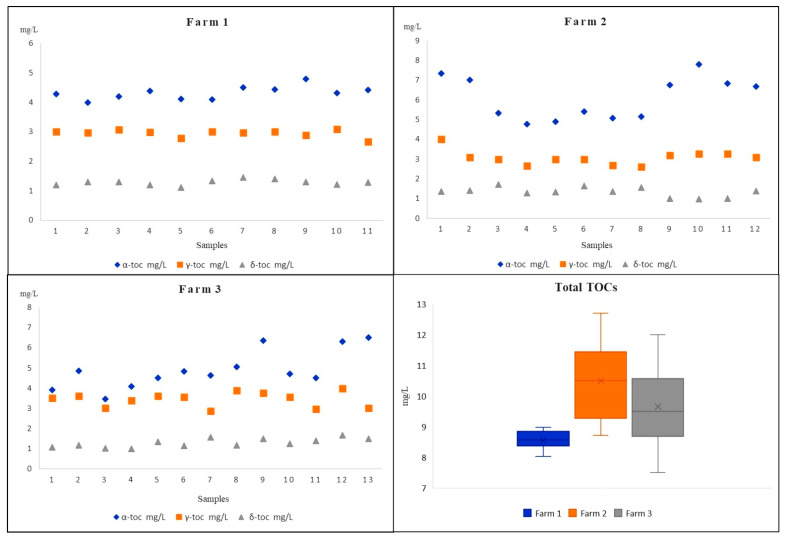
The TOCs’ content in different *Modicana* cow milks from three different farms.

**Table 1 molecules-26-04645-t001:** Analytical characteristic of the method.

Analyte	t_R_ ^1^ (min)	R^2^	LOD ^2^ (mg L^−1^)	LOQ ^2^ (mg L^−1^)	Level I		Level II		Level III	
					Recovery ^3^ (%)	RSD (%)	Recovery ^3^ (%)	RSD (%)	Recovery ^3^ (%)	RSD (%)
α-tocopherol	4.67	0.9986	0.030	0.101	90.5	5.2	91.8	3.5	95.3	0.4
γ-tocopherol	5.58	0.9987	0.027	0.089	88.6	5.9	87.8	3.3	91.6	0.3
δ-tocopherol	6.59	0.9983	0.029	0.097	88.4	7.4	89.8	3.6	92.8	0.6

^1^ t_R_, retention time; ^2^ LOD, limit of detection (3.3 σ/S); LOQ, limit of quantification (10 σ/S); ^3^ average of three replicates at different concentrations (level I = 0.5 mg L^−1^, level II = 2.5 mg L^−1^ and level III = 5 mg L^−1^).

**Table 2 molecules-26-04645-t002:** Precision parameters (expressed as relative standard deviation (RSD%)) obtained analyzing the standards mixture in optimized conditions.

Analyte	Intra-Day Precision (RSD%, *n* = 6)				Inter-Day Precision (RSD%, *n* = 12)			
	t_R_ ^1^	Area			t_R_ ^1^	Area		
		Level I (0.5 mg L^−1^)	Level II (2.5 mg L^−1^)	Level III (5 mg L^−1^)		Level I (0.5 mg L^−1^)	Level II (2.5 mg L^−1^)	Level III (5 mg L^−1^)
α-tocopherol	0.11	1.89	2.30	0.35	1.04	3.17	4.83	5.08
γ-tocopherol	0.13	2.25	1.09	0.57	1.40	5.20	5.14	3.95
δ-tocopherol	0.16	2.55	2.08	1.35	1.35	5.19	3.47	5.12

^1^ t_R_, retention time.

**Table 3 molecules-26-04645-t003:** Content (mg L^−1^) of tocopherols in milk samples from *Modicana* cattle breed.

	Analyte	Range	Mean ± SD	Median
*Farm A* (*n* = 11)	α-TC	4.00–4.79	4.32 ± 0.22	4.31
	γ-TC	2.66–3.09	2.95 ± 0.13	2.99
	δ-TC	1.12–1.46	1.29 ± 0.10	1.30
*Farm B* (*n* = 12)	α-TC	4.77–7.79	6.09 ± 1.08	6.05
	γ-TC	2.62–4.01	3.07 ± 0.37	3.05
	δ-TC	0.99–1.71	1.34 ± 0.24	1.37
*Farm C* (*n* = 13)	α-TC	3.46–6.50	4.91 ± 0.95	4.71
	γ-TC	2.86–3.99	3.45 ± 0.37	3.56
	δ-TC	1.01–1.68	1.30 ± 0.22	1.24

## Data Availability

Not applicable.

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
