# Peer review of "A Fast and Efficient Ultrasound-Assisted Extraction of Tocopherols in Cow Milk Followed by HPLC Determination"

_molecules, 2021, doi:10.3390/molecules26154645_

Round 1

Reviewer 1 Report

The authors describe a novel method of extraction and chromatographic determination of tocopherols in milk using HPLC coupled with fluorescence detection. This is good work with systematic, properly executed research.

However, some points should be corrected:

Abstract, "FD" should be explained. It is not clear.

Introduction: There is a lack of mass spectrometry detection

Line 129, I suggest using molar concentration. 

Fig. 1 It is possible to export data in better quality.

Table 2 Are the retention times in Table 2? The values look strange. 

Line 237 There is some mistake in spelling

Line 347, LabSolutions is a correct name.

Line 347 Please check the column and precolumn.

I do not understand the difference between Table 3 and Fig. 3. Data are rather the same. Why are they duplicated?

Author Response

Response to Reviewer 1 Comments

Point 1: Abstract, "FD" should be explained. It is not clear.

Response 1: “FD” was explained in abstract (Line 14).

Point 2: Introduction: There is a lack of mass spectrometry detection.

Response 2: Thanks for this remark; now two bibliographic references have been inserted. [23,30] (Line 106, Line 475-477, Line 495-497).

Point 3: Line 129, I suggest using molar concentration.

Response 3: Corrected (now Line 150).

Point 4: Fig. 1 It is possible to export data in better quality.

Response 4: The implementation of Fig.1 has been done.

Point 5: Table 2 Are the retention times in Table 2? The values look strange.

Response 5: The retention time are reported in Table 2 (second column of the table) and the data shown are provided directly from the HPLC program with the chromatogram.

Point 6: Line 237 There is some mistake in spelling

Response 6: The correction has been done (now Line 260).

Point 7: Line 347, LabSolutions is a correct name.

Response 7: The correction has been done (now Line 380).

Point 8: Line 347 Please check the column and precolumn.

Response 8: Thanks for this remark, unfortunately it was a typing error. Now the correction has been done (now Line 381).

Point 9: I do not understand the difference between Table 3 and Fig. 3. Data are rather the same. Why are they duplicated?

Response 9: According to the other reviewer comment we decided to keep both, Figure 3 showing the individual cows coming from each farm without graphic connection, and Table 3 which shows average values and ranges for the three different farms.

Reviewer 2 Report

The manuscript Rotondo et al. describes tocopherol detection following ultrasound-assisted  extraction from milk. The topic and proposed method are interesting and have merit for publication in Molecules.

General comments:

Introduction: There is a bit a mix when describing extraction related methods and HPLC detection, authors to better separate.

Provide some more specific details on temperature stability of TOC

Elaborate on drawbacks of saponification which is indicated.  

What is the TOC outcome from extractions without ultrasonication? Was this tested?

English language check required, a number of grammar and spelling mistakes. Quite a few of these have been highlighted and need to be applied throughout manuscript.  

Other comments:

Reword title as currently implies that extraction is part of HPLC method which is separate as HPLC is independent. 

Line 13, 14 "tocopherols determination" if combined with other expression, e.g determination, extraction, then tocopherol is singular (which has nothing to do with the emphasis of the manuscript on the different tocopherols). Apply throughout manuscript.    

L18 - rephrase to mention milk only has other matrices were not investigated. 

L30 – rephrase to remove “we”, instead use passive. Also place “expression improperly talk” by more scientific phrase (vitamin E is a group name, and the scientific community is clear about the presence of different tocopherols, and tocotrienols) and apply correct grammar.

L32 – compare rather than evaluate activity of other tocopherols

L38 – change to “TOCs exert strong antioxidant activities ..”

L39-43 – add the type of studies that is referred to, the currently cited are not in vivo studies, therefore some more details are required. And include moderation e.g. “may protect in vivo lipid peroxidation…

L57 – Tocopherols are mainly found….

L61 – not clear. Are authors implying a positive association of tocopherol content with milk quality? Rephrase to clarify.

L68 – comma after compounds to make it clear that interfering hydrophobic compounds refer to triglycerides.

L69 – strategies … TOC determination….

L72 – are authors referring to a specific method or typical approach, this needs to be clarified.

L74 – This is … again, in view of comments on previous sentence requires clarification

L87 – “a pretreatment step” or “pretreatment steps are”

L98 – replace surely by a more scientific expression

L100 – what is the reason for the lacking prediction of TOCs other than alpha-TOC? Content of these in comparison much lower?

L101 – addition of “and it is difficult ….” Not clear, as alpha-TOC was predicted

L103 – “aim of study was”

L114 – “was optimized” or “procedures … were”

L122 – rephrase e.g. to “commonly available..”. Remove “as unexpensive standard”, not necessary and, in view of HPLC, standard refers to compounds. (it would actually be inexpensive).

L124-126 – correct for English language and grammar errors

L134 – “no significant differences”

L137 – sentence not clear

L248 – briefly explain the approach of the Havermose method

L288 – Figure 3: Presumably the samples are from individual cows on each farm, why are the lines connected? Recommend to present data as box plots with dots for individual animals, which would also generate mean values for each TOC. Is it mentioned how often each sample was extracted and run on HPLC?

L303 – Authors should include some more details on the differences of TOC content from Modicana to other breeds milk (other than significantly rich – actually grammatically incorrect) as they have not analysis of other cow milk in their analyses.  

L339 – add optimized ultrasonic conditions in the methods part.

L371 – not convinced on this sentence with regards to quality of the milk without further reference (in discussion) to TOC content in conventional breeds.

L374 – rephrase last sentence. Very long, best to split into two. Better link the required further testing of TOC content with regards to factors such as seasonal variability, feeding regimes, lactation stage.

Author Response

Response to Reviewer 2 Comments

Point 1: General comments:

Introduction: There is a bit a mix when describing extraction related methods and HPLC detection, authors to better separate.

Provide some more specific details on temperature stability of TOC.

Elaborate on drawbacks of saponification which is indicated.  

What is the TOC outcome from extractions without ultrasonication? Was this tested?

English language check required, a number of grammar and spelling mistakes. Quite a few of these have been highlighted and need to be applied throughout manuscript.  

Response 1: Thank you for your comments which will allow us to greatly improve the quality of the manuscript. Therefore, following your indications we have separated the extraction techniques (Line 79-92) from the HPLC methods (Line 93-106).

As regards the need to perform an initial saponification on the substances to be analysed and the temperature stability of the tocopherols, we have included some bibliographical references [Ref. 23-26] and specified some precautions that are normally followed in alkaline saponification [Ref. 19,24].

The test on the tocopherol content at time zero, without ultrasonication, was not carried out as the extraction was not even complete after 5 minutes of sonication at 20 °C, as evidenced in Fig.1.

Finally, English language was checked throughout the manuscript.

Point 2: Reword title as currently implies that extraction is part of HPLC method which is separate as HPLC is independent. 

Response 2: The new title is “A fast and efficient ultrasound-assisted extraction of tocopherols in cow milk followed by HPLC determination”

Point 3: Line 13, 14 "tocopherols determination" if combined with other expression, e.g determination, extraction, then tocopherol is singular (which has nothing to do with the emphasis of the manuscript on the different tocopherols). Apply throughout manuscript.    

Response 3: We have edited expressions concerning TOCs throughout the paper hoping to match the reviewer suggestion.

Point 4: L18 - rephrase to mention milk only has other matrices were not investigated.

Response 4: The correction has been done.

Point 5: L30 – rephrase to remove “we”, instead use passive. Also place “expression improperly talk” by more scientific phrase (vitamin E is a group name, and the scientific community is clear about the presence of different tocopherols, and tocotrienols) and apply correct grammar.

Response 5: The text has been corrected.

Point 6: L32 – compare rather than evaluate activity of other tocopherols.

Response 6: The correction has been done (now Line 34).

Point 7: L38 – change to “TOCs exert strong antioxidant activities ..”

Response 7: The correction has been done (now Line 40).

Point 8: L39-43 – add the type of studies that is referred to, the currently cited are not in vivo studies, therefore some more details are required. And include moderation e.g. “may protect in vivo lipid peroxidation…

Response 8: Thanks for this remark. Now the correction has been done (Line: 41, 42, 44) and the new references [3,4] are:

  • Yoshida, Y.; Saito, Y.; Jones, L.S.; Shigeri, Y. Chemical reactivities and physical effects in comparison between tocopherols and tocotrienols: physiological significance and prospects as antioxidants. J. Biosci. Bioeng. 2007, 104, 439-445
  • Schneider, C. Chemistry and biology of vitamin E. Mol. Nutr. Food Res. 2005, 49, 7–30.

Point 9: L57 – Tocopherols are mainly found….

Response 9: The correction has been done (now Line 59).

Point 10: L61 – not clear. Are authors implying a positive association of tocopherol content with milk quality? Rephrase to clarify.

Response 10: We do not associate the milk quality with tocopherol content, therefore we used “composition” instead of “quality” (Line 64).

Point 11: L68 – comma after compounds to make it clear that interfering hydrophobic compounds refer to triglycerides.

Response 11: The correction has been done (now Line 71).

Point 12: L69 – strategies … TOC determination….

Response 12: The correction has been done and we specified that the study involved a‑tocopherol only (now Line 72).

Point 13: L72 – are authors referring to a specific method or typical approach, this needs to be clarified.

Response 13: It is a typical approach that are presented in different literature data; therefore now bibliographic references are added (Line 77).

Point 14: L74 – This is … again, in view of comments on previous sentence requires clarification

Response 14: We cited the literature [23,24] to support this sentence.

Point 15: L87 – “a pretreatment step” or “pretreatment steps are”

Response 15: The correction has been done (now Line 80).

Point 16: L98 – replace surely by a more scientific expression

Response 16: The correction has been done and we used “Certainly” (now Line 117).

Point 17: L100 – what is the reason for the lacking prediction of TOCs other than alpha-TOC? Content of these in comparison much lower?

Response 17: We arranged the paragraph and in order to avoid misunderstandings, we reported the exact words written by the authors of the study [Ref 16 = Soulat et al., 2020] (now Line 119-120).

Point 18: L101 – addition of “and it is difficult ….” Not clear, as alpha-TOC was predicted

Response 18: See answer point 17

Point 19: L103 – “aim of study was”

Response 19: The correction has been done (now Line 123).

Point 20: L114 – “was optimized” or “procedures … were”

Response 20: The correction has been done (now Line 134).

Point 21: L122 – rephrase e.g. to “commonly available..”. Remove “as unexpensive standard”, not necessary and, in view of HPLC, standard refers to compounds. (it would actually be inexpensive).

Response 21: The correction has been done (now Line 142).

Point 22: L124-126 – correct for English language and grammar errors

Response 22: The sentence has been written again. (Line 144-146)

Point 23: L134 – “no significant differences”

Response 23: The correction has been done (now Line 155).

Point 24: L137 – sentence not clear

Response 24: We modified the sentence with the aim to be clear.

The ultrasonic extraction can be studied by varying different parameters such as: ultrasonic power, ultrasonic frequency, ultrasonic time, and ultrasonic temperature. The device at our disposal did not allow to evaluate all these variables, therefore were carried out the study varying the sonication time and temperature only, since ultrasonic power and ultrasonic frequency were fixed.

Point 25: L248 – briefly explain the approach of the Havermose method

Response 25: The Havermose method was briefly explained (now Line 271,272).

Point 26: L288 – Figure 3: Presumably the samples are from individual cows on each farm, why are the lines connected? Recommend to present data as box plots with dots for individual animals, which would also generate mean values for each TOC. Is it mentioned how often each sample was extracted and run on HPLC?

Response 26: Figure 3 was modified following your indication. The quantitative analysis was carried out with the method of external standard using suitable calibration curves and any quantification was estimated as mean value of three repeated measurements. (Line 387-389).

Point 27: L303 – Authors should include some more details on the differences of TOC content from Modicana to other breeds milk (other than significantly rich – actually grammatically incorrect) as they have not analysis of other cow milk in their analyses.

Response 27: The paragraph was rephrased, and specific references [Ref 19, 42,43], and values were added according to the kind reviewer’s advice (now Line 320-327).

Point 28: L339 – add optimized ultrasonic conditions in the methods part.

Response 28: The correction has been done (now Line 372).

Point 29:

L371 – not convinced on this sentence with regards to quality of the milk without further reference (in discussion) to TOC content in conventional breeds.

L374 – rephrase last sentence. Very long, best to split into two. Better link the required further testing of TOC content with regards to factors such as seasonal variability, feeding regimes, lactation stage.

Response 29: Thanking the reviewer for the appropriate suggestion we have re-written the concluding paragraph (now Line 407-411).

Round 2

Reviewer 2 Report

The authors have addressed most issues flagged up earlier. 

The following comments: 

Line 134-136 : check grammar in this sentence. ....by performing different tests on a milk sample.....

Line 230 :  .... five point calibration curves were drawn in the range....

Line 245 : ...In an attempt...

Line 270 : ...All these data....

Line 414 : change part of last sentence (actually authors were asked to split the sentence), as the current version is too speculative and different emphasis as before. Remove last part (...lead to remarkable benefits for the human health). 

Author Response

Response to Reviewer 2 Comments

Point 1: Line 134-136: check grammar in this sentence. ....by performing different tests on a milk sample.....

Response 1: The sentence has been written again.

Point 2: Line 230:  .... five point calibration curves were drawn in the range....

Response 2: The correction has been done (now Line 214).

Point 3: Line 245: ...In an attempt...

Response 3: The correction has been done (now Line 229).

Point 4: Line 270: ...All these data....

Response 4: The correction has been done (now Line 254).

Point 5: Line 414: change part of last sentence (actually authors were asked to split the sentence), as the current version is too speculative and different emphasis as before. Remove last part (...lead to remarkable benefits for the human health).

Response 5: The text has been corrected. (now Line 382-386).